# Umbilical Cord Biometry and Fetal Abdominal Skinfold Assessment as Potential Biomarkers for Fetal Macrosomia in a Gestational Diabetes Romanian Cohort

**DOI:** 10.3390/medicina58091162

**Published:** 2022-08-26

**Authors:** Andreea Roxana Florian, Gheorghe Cruciat, Georgiana Nemeti, Adelina Staicu, Cristina Suciu, Mariam Chaikh Sulaiman, Iulian Goidescu, Daniel Muresan, Florin Stamatian

**Affiliations:** 1Obstetrics and Gynecology I, Mother and Child Department, University of Medicine and Pharmacy “Iuliu Hatieganu”, 400006 Cluj-Napoca, Romania; 2Imogen Clinical Research Centre, 400006 Cluj-Napoca, Romania

**Keywords:** gestational diabetes mellitus, fetal macrosomia, umbilical cord area, abdominal skinfold, Wharton jelly

## Abstract

*Background**and Objectives:* Gestational diabetes mellitus (GDM) is a pregnancy-associated pathology commonly resulting in macrosomic fetuses, a known culprit of obstetric complications. We aimed to evaluate the potential of umbilical cord biometry and fetal abdominal skinfold assessment as screening tools for fetal macrosomia in gestational diabetes mellitus pregnant women. *Materials and methods:* This was a prospective case–control study conducted on pregnant patients presenting at 24–28 weeks of gestation in a tertiary-level maternity hospital in Northern Romania. Fetal biometry, fetal weight estimation, umbilical cord area and circumference, areas of the umbilical vein and arteries, Wharton jelly (WJ) area and abdominal fold thickness measurements were performed. *Results:* A total of 51 patients were enrolled in the study, 26 patients in the GDM group and 25 patients in the non-GDM group. There was no evidence in favor of umbilical cord area and WJ amount assessments as predictors of fetal macrosomia (*p* > 0.05). However, there was a statistically significant difference in the abdominal skinfold measurement during the second trimester between macrosomic and normal-weight newborns in the GDM patient group (*p* = 0.016). The second-trimester abdominal circumference was statistically significantly correlated with fetal macrosomia at term in the GDM patient group with a p value of 0.003, as well as when considering the global prevalence of macrosomia in the studied populations, 0.001, when considering both populations. *Conclusions:* The measurements of cord and WJ could not be established as predictors of fetal macrosomia in our study populations, nor differentiate between pregnancies with and without GDM. Abdominal skinfold measurement and abdominal circumference measured during the second trimester may be important markers of fetal metabolic status in pregnancies complicated by GDM.

## 1. Introduction

Gestational diabetes mellitus (GDM) is defined as glucose intolerance newly diagnosed during pregnancy. Alongside type 1 and 2 diabetes mellitus, GDM prevalence has increased dramatically worldwide in the past decades [1]. Associated perinatal maternal–fetal consequences arise mostly from hyperglycemia per se but also due to complications such as excessive maternal weight gain, miscarriage, fetal anomalies, pre-eclampsia and fetal macrosomia.

Fetal macrosomia impacts both the fetal and maternal obstetric outcomes. Macrosomic fetuses have an increased risk of perinatal death, birth trauma due instrumental delivery and maneuvers required to address shoulder dystocia, neonatal hypoglycemia, hyperbilirubinemia, neonatal respiratory distress syndrome and longer neonatal intensive care unit admission intervals. In adulthood, these children are prone to developing impaired glucose tolerance, metabolic syndrome and cardiovascular diseases. Maternal effects range from complicated labor, higher rates of instrumental and operative deliveries and an increased risk of postpartum hemorrhage sometimes requiring transfusion of blood products to perineal trauma of different degrees, increased rates of caesarean deliveries and an augmented long-term risk of genital organ prolapse [2,3].

Despite the advances in ultrasound equipment technology, standards and guidance for fetal biometry measurements and better training of professionals, fetal macrosomia continues to represent a diagnostic issue. Overestimation of fetal weight leads to many unnecessary cesarean deliveries [4]. Contrarily, even correctly diagnosed, the outcome of vaginal delivery cannot be accurately predicted except for extremely macrosomic fetuses.

The umbilical cord (UC), which connects the fetus and placenta, has been extensively studied in recent years, anatomically, morphologically and by ultrasonography. It contains the umbilical vessels (normally, two umbilical arteries and one vein) and the remnant of the allantois, embedded in Wharton’s jelly (WJ)—a network of glycoprotein microfibrils, collagen fibers and hyaluronic acid—surrounded by a single layer of amnion [5]. Wharton’s jelly ensures a normal blood flow through the umbilical cord, preventing the collapse or knotting of the cord and therefore the disruption of vascular flow through the blood vessels. Various umbilical cord anomalies in size, vessel number, course and connection, structure and configuration have been described, with sometimes no impact on the course of pregnancy, but other times associated with various pregnancy conditions.

The umbilical cord is a vital structure for fetal development, and its detailed analysis can provide valuable information to allow the estimation of neonatal outcomes in various pregnancy-related pathologies. Changes in the amount of Wharton’s jelly have been linked to the occurrence of pregnancy-associated pathologies such as pregnancy-induced hypertensive disease, gestational diabetes mellitus and stillbirth [6]. Decreases in WJ quantity, changes in its protein structure and variations in the size of the umbilical vessel area have been associated with the development of pre-eclampsia [7,8]. The link between increased umbilical cord diameter and the development of GDM and the relationship between IUGR and the reduced cord diameter with decreased WJ are subjects of debate [7,9,10,11]. An increased diameter of the UC has been found in cases of fetal macrosomia and aneuploidy [12,13,14].

The main objective of our study was to determine whether the ultrasound measurement of umbilical cord anthropometric parameters (umbilical cord area, umbilical cord vessel area, the amount of Wharton’s jelly) and fetal abdominal skinfold assessment can be used as diagnostic or prognostic tools for fetal macrosomia in a selected population of Romanian patients.

## 2. Materials and Methods

This was a prospective case–control study conducted between January 2021 and June 2021 in the Obstetrics-Gynecology I Outpatient Department of the Emergency Cluj-Napoca County Hospital, Romania.

All procedures performed were in accordance with the national and European legislation (Declaration of Helsinki 1964), the enrollment of patients being carried out following counseling and obtaining their informed consent.

Inclusion criteria for the study group were represented by pregnant women with singleton pregnancies, during 24–28 weeks of gestation (WG), whose pregnancies were monitored in the Outpatient Department of the Emergency Cluj-Napoca County Hospital, with a planned delivery in the Obstetrics and Gynecology I maternity hospital.

The following represented study exclusion criteria: pre-existing type 1 or type 2 diabetes mellitus, the coexistence of pregnancy-associated pathologies, single umbilical artery, patients whose pregnancies were not monitored in the Outpatient Department of the ECCN, patient refusal to participate in the study.

All patients were evaluated and examined at the 24–28 WG pregnancy follow-up visit. Upon presentation, they were counseled about the check-up protocol and study implications and accepted or declined enrollment. Gestational diabetes mellitus screening by oral glucose tolerance test (OGTT) was performed in the morning, followed by ultrasound as the second step of evaluation. Patient family, medical, surgical and obstetric history, as well as current pregnancy history, maternal and fetal outcome data, were collected from medical records.

Screening for GDM was carried out according to IADPSG (International Association of the Diabetes and Pregnancy Study Groups) recommendations, by performing the OGTT with a 75 mg glucose load and 3 glycemia measurements: fasting, one hour and two hours following glucose ingestion. Gestational diabetes was diagnosed when one of the three values taken was altered: fasting blood glucose > 92 mg/dL (5.1 mmol/L), 1 h glycemia level > 180 mg/dL (10 mmol/L), 2 h glycemia level > 153 mg/dL (8.5 mmol/L) [15]

Fetal biometry, fetal weight estimation based on the formula C of Hadlock et al. (Hadlock C; log(10) BW = 1.335 − 0.0034(abdominal circumference (AC))(femur length (FL)) + 0.0316(biparietal diameter) + 0.0457(AC) + 0.1623(FL)] [16] umbilical cord area and circumference, areas of the umbilical vein and arteries, WJ area and abdominal fold thickness measurements were performed in all patients enrolled in the study by two examiners specializing in maternal–fetal medicine who performed examinations alternatively. All measurements were carried out using a GE VolusonE8 Expert with RAB6-D, 2–7 MHz convex abdominal probe. Ultrasonographers were blinded regarding the diagnosis of GDM (OGTT test results) in the respective pregnancy so as not to influence measurements.

Umbilical cord measurements were performed at the level of a free cord loop no more than 2 cm away from the fetal abdominal wall [9]. The umbilical cord diameter was meared circumferentially, outside the umbilical cord, using the “ellipse” measurement function of the ultrasound machine followed by the automatic calculation of the measured area (Figure 1A). Cord vessels and vessel area were similarly measured. The Wharton jelly area was calculated by subtracting the areas of the umbilical vein and arteries from the entire cross-sectional area of the cord. Fetal abdominal skinfold was identified as an external hyperechogenic surface on the standard transverse plane for the assessment of the abdominal circumference, the level at which measurements were performed. Fold thickness was measured by placing one caliper precisely between the fetal skin and adjacent amniotic fluid, with the second caliper being placed between the subcutaneous fat layer and the anterior abdominal wall (Figure 1B) [17].

Macrosomia was defined as estimated fetal weight above the 95th centile for any gestational age, at any point during pregnancy, or birth weight equal to or more than 4000 g [18].

### Statistical Analysis

Analysis was performed using MedCalc Statistical Software version 19.1.5 (MedCalc Software, Ostend, Belgium; https://www.medcalc.org; Accessed on 25 May 2020). Continuous variables were tested for normality of distribution (Shapiro–Wilk test) and were described by means ± standard deviation. Nominal data are characterized by means of frequencies and percentages. Comparisons between groups were performed using the Mann–Whitney or chi-square test, whenever appropriate. The model included the variables that achieved a *p* value < 0.05 in the univariate analysis. A *p* value < 0.05 was considered statistically significant.

## 3. Results

The study group consisted of 51 pregnant patients who presented for the 24–28 WG follow-up visit and agreed to take part in the study. According to the OGTT test results, patients were divided into two groups, the GDM patient group (26 patients) and the non-GDM group, consisting of 25 patients.

The demographic characteristics of the study population are presented in Table 1.

Biometry evaluation for estimated fetal weight, as well as corresponding centiles for gestational age, measurement of UC parameters and abdominal skinfold assessment comparative results for the two groups are presented in Table 2.

The characteristics of the study group patients from the perspective of obstetric outcome, delivery and neonatal weight are presented comparatively in Table 3.

When considering the parity and route of delivery, in the GDM study group, most patients were multiparous and gave birth via C-section.

The caesarean section indications were previous caesarean delivery (37.5% GDM, 42.8% non-GDM group), failed induction (25% GDM), breech presentation (6.25% GDM; 14.28 non-GDM), maternal pathology (31.25% GDM; 14.28% non-GDM) and placental insufficiency (21.42% non-GDM).

Table 4 depicts the correlation between fetal ultrasound parameters and maternal weight gain during pregnancy and term macrosomia.

In the GDM patient group, there was a statistically significant correlation between the second-trimester fetal weight and birthweight (*p* = 0.012), while no such correlation could be established in the non-GDM pregnancy group.

Umbilical cord area and WJ amount assessment could not be established as predictors of fetal macrosomia.

However, there was a statistically significant difference between the abdominal skinfold measurement during the second trimester between macrosomic and normal-weight newborns in the GDM patient group (*p* = 0.016). This confirms abdominal skinfold measurement during the second-trimester ultrasound evaluation as a potential predictor of fetal macrosomia at term in GDM pregnancies.

There was a statistically significant difference between the macrosomic and normal weighted fetuses at term in the GDM patient group when considering the second-trimester estimated fetal weight (*p* = 0.000), but also when considering the global prevalence of macrosomia in the studied populations (*p* = 0.001). The same was true for the measurement of the abdominal circumference during the second trimester with a *p* value of 0.003 for the GDM group, respectively 0.001 when considering both populations.

We also found a statistically significant difference between the fetal macrosomia status at term regarding the maternal weight gain during pregnancy when calculating for both patient groups globally (*p* = 0.010), but not when considering only the GDM patient sample (*p* = 0.643).

## 4. Discussion

The delivery of a macrosomic neonate leads to important maternal–fetal short- and long-term consequences. Diagnosis and prevention of fetal macrosomia thus represent obstetric priorities for reducing maternal and fetal morbidity. Several studies have suggested that ultrasound evaluation of umbilical cord biometry parameters and abdominal skinfold measurement could be a powerful adjunct in the prediction of term macrosomia [18,19]. Our purpose was to test the utility of these measurements as predictive and diagnostic tools for fetal macrosomia in a Romanian population.

Measurement of umbilical cord parameters at the time of GDM screening, prior to the initiation of dietary and therapeutic interventions, was targeted in our research. Ultrasound measurements of the umbilical cord parameters and fetal abdomen showed higher values in fetuses from mothers diagnosed with GDM, such as umbilical cord area, WJ area, abdominal circumference and abdominal skinfold. Umbilical cord area and Wharton jelly could not be established as predictors of fetal macrosomia in our study. Abdominal skinfold and the abdominal circumference measurements in the second trimester were significantly higher in the fetuses from GDM pregnancies compared to fetuses from non-GDM pregnancies. The fetal abdominal skinfold, the abdominal circumference, estimated second-trimester fetal weight and maternal weight gain during pregnancy were also statistically significantly correlated with fetal macrosomia at term.

It has been reported that umbilical cord cross-sectional measurement and estimation of umbilical cord components correlate with fetal size [20]. Our study demonstrated a statistically significant relationship between estimated second-trimester fetal weight, maternal weight gain during pregnancy and macrosomia at term/delivery.

A study conducted by Pietryga et al. evaluated umbilical cord and WJ areas in GDM and non-GDM patients between 22 and 40 WG and concluded that there were no differences between the parameters measured in the two groups and supported the limited value of cord biometry as a predictor of macrosomia at birth in GDM patients [21].

In our study, we found no statistically significant correlations between umbilical cord area umbilical cord area, the amount of cross-sectional WJ and birthweight, respectively, in GDM and non-GDM patients. The explanation for this may be that in pregnancies with GDM, treatment and diet applied after diagnosis prevent the development of fetal macrosomia as well as the umbilical cord and WJ biometry. This conclusion was also reached by Naylor et al., who showed that fetal birthweight is normal in GDM pregnancies because of timely and proper treatment [22].

Structural or volume changes in the umbilical cord and the amount of Wharton’s jelly have been shown to signal unfavorable maternal and fetal outcomes. These parameters may provide information on the efficiency of management in various pregnancy-related pathologies, such as pre-eclampsia, intrauterine growth restriction, stillbirth and intrapartum fetal distress [14]. Based on these premises, attempts have been made to incorporate cord area measurement into fetal biometry assessment to improve detection of either macrosomia or growth-restricted fetuses. Nomograms for gestational-age specific cord parameter values have been calculated. It appears that umbilical cord area and WJ area gradually increase up to 30 WG, reach a maximum value around 34 WG and then remain constant [20,23,24].

Weissman et al. confirmed the finding of increased cord areas in pregnancies complicated by GDM on account of increased WJ measurements [25]. This is presumably due to hemorrhage in the umbilical artery walls, increased permeability and subsequent plasma extravasation [26]. The authors speculated that differences in WJ may be useful to discriminate between constitutional macrosomia from macrosomia in GDM-complicated pregnancies. However, in our study, there was no statistically significant difference between the measurement of cord area and WJ between GDM and non-GDM pregnancies.

Several authors obtained similar results to ours regarding the correlation between fetal weight at 24–28 WG and birthweight. A similar highly significant correlation between fetal weight estimation at 26–28 WG and neonatal macrosomia at delivery was recorded by Togni et al. This comes as proof of the correspondence between fetal biometry estimation before 33–34 WG, the amount of WJ and excessive fetal weight at birth. It was also postulated that, after 34 WG, the amount of WJ decreases progressively and its measurement is no longer useful in estimating fetal weight [27]. A more recent study assessed the value of abdominal circumference and estimated fetal weight growth velocity in the prediction of macrosomia but established they do not improve detection as compared to the standard third trimester estimation using the Hadlock formula [28]. The amount of fat in the fetal body and therefore the thickness of the abdominal fold is influenced by numerous factors, such as race, maternal BMI, maternal weight gain during pregnancy, family factors, maternal hypertensive pathology or GDM [17,29,30].

Maternal hyperglycemia and subsequent fetal hyperglycemia lead to hyperactivation of the pancreas with excessive insulin production. The endpoint of these physiopathologic processes is a macrosomic fetus due to increased fat and protein stores.

Jain et al. found a highly significant statistical correlation between serial fetal abdominal circumference measurements at 30–32 and 36–38 WG, respectively, and increased birth weight. In the same study, statistically significant differences were recorded at 30–32 WG between fetal abdominal circumference values in GDM and non-GDM patients [31].

The finding of increased fetal abdominal skinfold in GDM patients resulting from our research was also reported by de Santis et al., who noted an exponential increase with gestational age and a higher developmental curve in fetuses from mothers with GDM compared to control fetuses from non-GDM pregnancies [32]. Bernstein et al. studied changes in fetal fat mass during pregnancy and found that it increased approximately tenfold between 19 and 40 WG [33].

An increased pre-pregnancy BMI and excessive maternal weight gain in pregnancy have been recognized as prerequisites for fetal macrosomia by many researchers [34,35,36,37]. Even a protective effect of low pre-pregnancy BMI in non-diabetic women has been described [38,39]. In our study groups, maternal pre-pregnancy obesity did not correlate with macrosomia, while excessive pregnancy weight gain correlated with large fetuses only when considering both groups and not the GDM category solely. This might be due to the limited patient number studied.

The delivery route of GDM patients in our study group was clearly balanced towards operative deliveries. Caesarean section accounted for 61.5% of the total number of deliveries, similar to the value of 73.2% reported by Naylor, and this occurred when not all the fetuses were macrosomic [22]. The high percentage of caesarean section deliveries, even with a normal weight fetus, was due to the large number of patients with obstetric indications, failed induction and patient refusal of induction.

There are several limitations to our study. Firstly, the size of the study groups, which does not allow the generalization of results. Another potential limitation/aspect is related to the technique of measuring cord parameters at the level of a single free cord loop. As this is not yet standardized, further studies are needed to establish the optimal number and points of measurement, considering that the thickness of the umbilical cord may vary along its length. Further research on larger groups of patients could help clinicians to achieve a better selection of patients at risk of developing GDM and may open new horizons in terms of early therapeutic management. Large-scale studies are needed to assess the clinical value of including fetal adnexal measurements in fetal weight estimation formulas. This is an easily reproducible technique, with a short learning curve, which could easily be implemented as a screening tool for fetal macrosomia. Another limitation of the current study is the lack of neonatal follow-up, which could allow the assessment of postpartum complications. At the same time, a more detailed study regarding the prediction of macrosomia might include the evaluation of maternal anthropometric parameters and a second cord biometry measurement at term. The adjunct of these parameters could enhance macrosomia detection.

## 5. Conclusions

In conclusion, in our study populations, the measurements of the cord and WJ could not be established as predictors of fetal macrosomia, nor differentiate between pregnancies with and without GDM. On the other hand, fetal abdominal skinfold measurement, fetal abdominal circumference, estimated second-trimester fetal weight and maternal weight gain during pregnancy may be important markers of fetal metabolic status in pregnancies complicated by GDM.

## Figures and Tables

**Figure 1 medicina-58-01162-f001:**
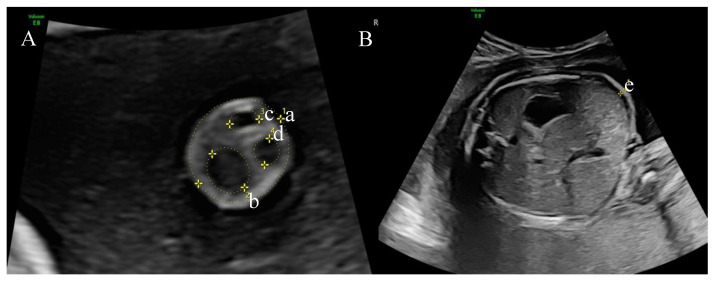
(**A**)**.** Umbilical cord biometry ultrasound landmarks: a. umbilical cord area; b. umbilical cord vein; c,d. umbilical cord artery; (**B**). Fetal transverse abdominal plane: e. fetal abdominal skinfold.

**Table 1 medicina-58-01162-t001:** Demographic data of the study populations.

	GDM Patients Study Group (*n* = 26)	Non-GDM Patients Study Group (*n* = 25)	*p*-Value
	arithmetic mean ± SD *		
Age (years)	32.5 ± 4.95	29.96 ± 3.72	0.044
Pre-gestational BMI	23.23 ± 4.00	22.22 ± 3.41	0.340
Pregnancy weight gain (kg)	13.35 ± 4.65	14.52 ± 5.26	0.402
Final BMI	28.13 ± 4.13	27.71 ± 4	0.718
	number (%)		
Provenance ** Urban area Rural area	20 (76.9%) 6 (23.1%)	17 (68.0%) 8 (32.0%)	0.689
Family history of diabetes mellitus	5 (19.23%)	3 (12%)	0.703
Smoking habit	3 (11.5%)	4 (16%)	0.703

* Standard deviation. ** According to the Organization for Economic Cooperation and Development (OECD)—OECD Regional Outlook 2016—Productive Regions for Inclusive Societies, OECD Publishing, Paris, https://doi.org/10.1787/9789264260245-en (accessed on 27 July 2022).

**Table 2 medicina-58-01162-t002:** Fetal biometry, umbilical cord and abdominal skinfold measurement of the study populations performed during the second trimester.

	GDM Patients Study Group (*n* = 26)	Non-GDM Patients Study Group (*n* = 25)	*p* Value
Fetal estimated weight 2nd trimester (g)	951.96 ± 145.08	975.96 ± 233.07	0.659
Fetal weight 2nd trimester (centiles)	59.36 ± 11.44	55.92 ± 10.31	0.265
Umbilical cord area (cm^2^)	2.03 ± 0.48	1.86 ± 0.41	0.189
Umbilical cord circumference (cm)	5.05 ± 0.66	4.82 ± 0.57	0.201
Umbilical cord vein (cm^2^)	0.37 ± 0.13	0.32 ± 0.11	0.121
Umbilical cord artery 1 (cm^2^)	0.09 ± 0.11	0.07 ± 0.02	0.363
Umbilical cord artery 2 (cm^2^)	0.09 ± 0.13	0.07 ± 0.02	0.329
Wharton jelly (cm^2^)	1.46 ± 0.42	1.39 ± 0.35	0.517
Abdominal skin fold (cm)	0.32 ± 0.07	0.22 ± 0.07	0.000
Abdominal circumference (cm)	22.14 ± 1.28	22.16 ± 1.93	0.970
Abdominal circumference (%)	60.18 ± 23.74	53.76 ± 23.53	0.337

**Table 3 medicina-58-01162-t003:** Pregnancy outcome parameters of the study group patients compared to the control group patients.

	GDM Patients Study Group (*n* = 26)	Non-GDM Patients Study Group (*n* = 25)	*p* Value
	number (%)		
Parity			
Nulliparous Multiparous	12 (46.2%) 14 (54.8%)	15 (60%) 10 (40%)	0.605
Route of delivery			
Vaginal C-section	10 (38.5%) 16 (61.5%)	11 (44%) 14 (56%)	0.907
	arithmetic mean ± SD *		
Birthweight (g)	3487.31 ± 435.75	3388 ± 548.54	0.477
Birthweight (centiles)	74.62 ± 22.89	71.72 ± 24.72	0.666
Macrosomia (>95 centile)	8 (30.8%)	8 (32%)	1.000

* standard deviation.

**Table 4 medicina-58-01162-t004:** The correlations of fetal ultrasound measurements during the second trimester and fetal macrosomia in both the entire group and the GDM population sample.

	Mean ± SD *		*p* Value
Correlations in the entire population sample (51 patients)			
2nd-trimester estimated fetal weight—birthweight (%)			0.007
2nd-trimester estimated fetal weight (%)	57.67 ± 1.53		
Birthweight (%)	73.20 ± 3.31		
	Normal weight fetuses (*n* = 35)	Macrosomic fetuses (*n* = 16)	
Abdominal skinfold 2nd (cm)—term macrosomia	0.256 ± 0.012	0.295 ± 0.028	0.135
2nd-trimester estimated fetal weight (%)—term macrosomia	54.25 ± 1.63	65.16 ± 2.51	0.001
2nd-trimester abdominal circumference(%)—term macrosomia	49.83 ± 3.84	72.81 ± 4.34	0.001
Excessive maternal weight gain (kg)—term macrosomia	12.74 ± 0.69	16.50 ± 1.44	0.010
Correlations in the GDM group (26 patients)			
2nd-trimester estimated fetal weight—birthweight (%)			0.012
2nd-trimester estimated fetal weight (%)	59.36 ± 2.24		
Birthweight (%)	74.62 ± 4.49		
	Normal-weight fetuses (*n* = 18)	Macrosomic fetuses (*n* = 8)	
Abdominal skinfold 2nd-trimester (cm)—term macrosomia	0.269 ± 0.011	0.369 ± 0.034	0.016
2nd-estimated fetal weight (%)term macrosomia	54.52 ± 2.36	70.25 ± 1.91	0.000
2nd-trimester abdominal circumference (%)—term macrosomia	51.58 ± 5.18	79.55 ± 5.25	0.003
Excessive maternal weight gain (kg)—term macrosomia	13.06 ± 1.07	14.00 ± 1.18	0.643

* standard deviation.

## Data Availability

The data presented in this study are available on request from the corresponding author. The data are not publicly available being part of the ECCN Cluj-Napoca, Romania archives.

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
