# Peer review of "Umbilical Cord Biometry and Fetal Abdominal Skinfold Assessment as Potential Biomarkers for Fetal Macrosomia in a Gestational Diabetes Romanian Cohort"

_medicina, 2022, doi:10.3390/medicina58091162_

Round 1
Reviewer 1 Report
This paper deals with whether the ultrasound measurement of the structures in umbilical cord or fetal abdominal skinfold can be diagnostic or prognostic tools for fetal macrosomia in some population of Romanian patients. This topic is interesting and deserves a constructive discussion. The authors contended that the measurement of cord and WJ could not be predictors of fetal macrosomia and abdominal circumference measured during the second trimester may be important markers of fetal metabolic status in pregnancies complicated by GDM.
However, I concern some issues as listed below for publication.
I hope that my comment is useful for the improvement of the article.
Major concerns:
1. Some P values are shown in the results, but some of them are not shown in Table 1-3. The authors need to present the quality of these data. In other words, the size, the mean parameter value, and the error of the 2 groups from which the P value is derived should be shown.
2. The authors has evaluated the route of delivery and the birthweight, but it would be of interest to the readers to further evaluate other outcomes, such as the occurrence of hyperglycemia in the newborn after delivery.
Minor concerns:
1. The authors should define the term “Urban area” and “Rural area” in the Table 1.
2. In line 171, “the was a statistically significant…” is probably a mistake for “there was a statistically significant…”.
3. In the results, there is a mix of notations with and without spaces before and after the equal. Please unify the notation
Reviewer 2 Report
Dear authors,
We have read the study with interest and we find several flaws. We do not agree that this is a case-control study. The design of the study must be reviewed. Perhaps you mean a retrospective cohort study. Anyway, the sample is quite small to accomplish the aims.
Do you think that the study has enough statical power to disaggregate the sample into two groups? In my opinion, as a first step, you should consider studying the association of these measurements with the dependent variables in the whole group. Otherwise, the sample size should be much higher.
Hadlock has several formulas for birthweight calculation. Which one did you use in this study?
At which gestational age did you perform the measurements? Were they standard medical controls? In that case, how could you reassure the blindness of the measurements?
How did you standardize the skinfold measurement?
How many researchers performed the measurements? Did you study the reproductivity of these measurements by the means of the kappa coefficient for example?
What´s the explanation for the high rate of c-sections in both groups?
What was the value of the correlation coefficients?
Round 2
Reviewer 1 Report
I think the manuscript has been revised well. I think this manuscript will be acceptable after some corrections have been done.
1. What are the units for the values listed in Table 4? The authors need to state that.
2. There seems to be a shaky notation in some of the Tables. For example, second and 2nd, Standard deviation and St dev, etc. To avoid confusion, the notation should be unified.
Author Response
Response to Reviewer 1
Esteemed Reviewer,
Thank you again for helping us upgrade the quality of our paper with your observations.
Reviewer - What are the units for the values listed in Table 4? The authors need to state that.
Authors –
Reviewer - There seems to be a shaky notation in some of the Tables. For example, second and 2nd, Standard deviation and St dev, etc. To avoid confusion, the notation should be unified.
Authors – We have unified the notations in all tables, we apologize for being careless and thank you for the observation.
Reviewer 2 Report
The authors have included some changes that increased the quality of the paper. However, Table 4 needs revision because we do not find sense in the figures. Are you reporting correlation indexes mean weight values or mean weight differences...? You also have to include units (Kg or gr...).
On the other hand, you should not name the "physiologic group" to the non-GDM group since there were more than 14% of non-diabetic pregnant women with other maternal conditions.
Author Response
Response for reviewer 2
Esteemed Reviewer
We highly appreciate your effort and support to help us shape up our manuscript.
Reviewer - The authors have included some changes that increased the quality of the paper. However, Table 4 needs revision because we do not find sense in the figures. Are you reporting correlation indexes mean weight values or mean weight differences...? You also have to include units (Kg or gr...).
Authors – Please do not consider us too thick, we do not understand. We introduced Table 4 in the manuscript following certain requirements made by the other reviewer in order to explain quality of the data. In other words, the size, the mean parameter value, and the error of the 2 groups from which the P value was derived.
As we see it, table 4 reports our calculation of the statistical power of correlations between qualitative and quantitative variables. For quantitative data (such as second trimester parameters and birthweight) we calculated the correlation between these variables per se.
In the attempt to mend things, we changed the phrasing in the Results section as follows: “There was a statistically significant difference between the macrosomic and normal weighted fetuses at term in the GDM patient group when considering the second trimester estimated fetal weight (p = 0.000), but also when considering the global prevalence of macrosomia in the studied populations (p = 0.001). The same was true for the measurement of the abdominal circumference during the second trimester with a p value of 0.003 for the GDM group, respectively 0.001 when considering both populations.
We also found a statistically significant difference between the fetal macrosomia status at term regarding the maternal weight gain during pregnancy when calculating for both patient groups globally (p = 0.010), but not when considering only the GDM patient sample (p = 0.643).”
I am not very sure we managed to answer your question. We are fully amenable to making other changes if required. Thank you.
Reviewer - On the other hand, you should not name the "physiologic group" to the non-GDM group since there were more than 14% of non-diabetic pregnant women with other maternal conditions.
Authors – Thank you for pointing this out, you are very right, and we have changed the phrasing everywhere to “non GDM group”.